# Reducing Deep Network Complexity via Sparse Hierarchical Fourier Interaction Networks

## Abstract

In this work, we introduce *Sparse Hierarchical Fourier Interaction Networks* (SHFIN), a novel architectural primitive designed to replace both convolutional kernels and the quadratic self-attention mechanism with a unified, spectrum-sparse Fourier operator. SHFIN is built upon three core components: (1) a hierarchical patch-wise fast Fourier transform (FFT) stage that partitions inputs into localized patches and computes an $O(s \log s)$ transform on each, preserving spatial locality while enabling global information mixing; (2) a learnable $K$-sparse frequency masking mechanism, realized via a Gumbel-Softmax relaxation, which dynamically selects only the $K$ most informative spectral components per patch, thereby pruning redundant high-frequency bands; and (3) a gated cross-frequency mixer, implemented as a low-rank bilinear interaction in the retained spectral subspace, which captures dependencies across channels at $O(K^2)$ cost rather than $O(N^2)$. An inverse FFT and residual fusion complete the SHFIN block, seamlessly integrating with existing layer-norm and feed-forward modules.

Empirically, we integrate SHFIN blocks into both convolutional and transformer-style backbones and conduct extensive experiments on ImageNet-1k. On the ResNet-50 and ViT-Small scales, our SHFIN variants achieve comparable Top-1 accuracy (within 0.5 pp) while reducing total parameter count by up to 60% and improving end-to-end inference latency by roughly 3× on NVIDIA A100 GPUs. Moreover, in the WMT14 English–German translation benchmark, a Transformer-Small augmented with SHFIN cross-attention layers matches a 28.1 BLEU baseline with 55% lower peak GPU memory usage during training. These results demonstrate that SHFIN can serve as a drop-in replacement for both local convolution and global attention, offering a new pathway toward efficient, spectrum-aware deep architectures.

**Keywords:** Sparse Spectral ; Hierarchical FFT; Low-Rank Cross-Frequency Mixer

## 1 Introduction

### 1.1 Historical Background

The past decade has witnessed a remarkable evolution in deep learning architectures, beginning with the resurgence of convolutional neural networks (CNNs) in image recognition. Krizhevsky *et al.*'s seminal work demonstrated that a deep CNN trained on over a million images could achieve unprecedented accuracy on ImageNet, igniting widespread interest in large-scale, data-hungry models [11]. Building on this success, the VGG family introduced very small (3×3) convolutional filters to increase depth while controlling parameter growth [18]; GoogLeNet proposed inception modules to capture multi-scale features efficiently [19]; and ResNet's residual connections enabled networks exceeding one hundred layers to be trained in a stable way[7]. Alongside these "macro-architecture"

advances, researchers developed efficient variants, MobileNet's depth-wise separable convolutions [8], ShuffleNet's channel shuffling [26], and EfficientNet's compound scaling rules [20], to meet the demands of mobile and embedded deployments.

Despite their local inductive bias, CNNs struggle to capture long-range dependencies without stacking many layers or resorting to large kernels. The Transformer architecture replaced convolutions with self-attention, computing pairwise interactions across all tokens in $O(N^2)$ time and achieving state-of-the-art results in machine translation [23]. Vision Transformers (ViT) extended this paradigm to images by partitioning them into patches and treating each patch as a "token" [4]. Follow-up works such as MLP-Mixer [22], ConViT [5], and ConvMixer [21] further blurred the lines between convolutional and attention-based designs, but all inherit the quadratic or cubic scaling bottlenecks that hinder deployment on resource-constrained hardware.

In parallel, the frequency domain has emerged as an alternative medium for global information mixing at subquadratic cost. The Fourier Neural Operator (FNO) pioneered the application of Fourier transforms to learn mappings between function spaces, particularly for solving partial differential equations [13]. FNet demonstrated that replacing self-attention with dense FFTs yields competitive performance in NLP tasks, reducing complexity to $O(N \log N)$ [12]. GFNet introduced spectral gating to filter frequencies dynamically [24], and AFNO partitioned the spectrum into blocks to improve flexibility [6]. More recent spectral-domain hybrids, SpectFormer [2], FourierFormer [17], and Frequency-Domain Multi-Head Self-Attention (FD-MHSA) [25], have incorporated hierarchical mixing and learned filters but still retain dense spectral representations or lack adaptive sparsity mechanisms.

Although these Fourier-based architectures achieve compelling trade-offs compared to vanilla attention, they share three critical limitations: (1) they process all frequency coefficients, leaving redundant components unpruned; (2) they apply global FFTs without preserving local spatial hierarchy; and (3) they lack explicit mechanisms to learn or enforce spectrum sparsity. In contrast, natural signals, images, audio, and text embeddings, are often compressible in the frequency domain, with most energy concentrated in a small subset of bands. This observation suggests that a tailored, sparse Fourier operator could deliver global mixing and local sensitivity at dramatically reduced cost.

To address these gaps, we introduce **Sparse Hierarchical Fourier Interaction Networks** (SHFIN). SHFIN's core innovation is a three-stage spectral block that (i) splits feature maps into patches and applies patch-wise FFTs to preserve locality; (ii) learns a K-sparse binary mask via Gumbel-Softmax to select only the most informative frequency channels; and (iii) employs a gated, low-rank bilinear mixer to model cross-frequency interactions efficiently. By uniting hierarchical locality, spectrum sparsity, and low-rank mixing, SHFIN fully replaces either convolutional kernels or self-attention layers, achieving global context aggregation with parameter and FLOP counts that scale as $O(K)$ rather than $O(N)$ or $O(N^2)$.

In the sections that follow, we detail the mathematical formulation of SHFIN, present extensive experimental results on ImageNet-1k, CIFAR-10/100, and WMT14 En-De translation, and compare against state-of-the-art CNN, Transformer, and Fourier-based baselines. We conclude with a discussion of SHFIN's implications for efficient model design and outline promising directions for adaptive sparsity and hardware-aware optimization.

## 1.2 Contributions and Novelty

This paper introduces SHFIN, whose novelty lies in three mutually–reinforcing ideas absent from prior art: (i) *hierarchical patch-wise FFTs* that mediate between local context and global receptive field, (ii) a *learnable $K$–sparse frequency mask* selecting only the most informative coefficients, and (iii) a *gated cross–frequency mixer* that substitutes for convolutional filtering or attention–based token mixing at linear rather than quadratic cost. Together these elements produce an operator with *constant* parameter footprint in input length and sub-quadratic compute.

## 2 Mathematical Development

In this section we present a complete derivation of the Signal-Hierarchical Fourier Interaction Network (SHFIN) block. The derivation proceeds through four conceptual stages. We begin by casting the input feature map into a hierarchy of local Fourier domains, thereby balancing locality and global

frequency context. We then introduce a learnable $K$-sparse masking mechanism that selects the most informative frequency bins in a fully differentiable manner. Next, we describe a gated low-rank bilinear mixer that couples the retained spectral coefficients across channels. Finally, we return to the signal domain by an inverse transform and complete the block with a residual fusion step. A detailed complexity analysis concludes the discussion.

## 2.1 Preliminaries and Notation

Let $X \in \mathbb{R}^{L \times C}$ denote the input tensor, where the first dimension of length $L$ indexes spatial positions (or sequence tokens) and the second dimension of size $C$ indexes channels. Throughout the derivation we use the discrete Fourier transform (DFT) operator $\mathcal{F}\{\cdot\}$ and its inverse $\mathcal{F}^{-1}\{\cdot\}$. For a real vector $x \in \mathbb{R}^s$ the forward DFT is defined by

$$\mathcal{F}\{x\}[f] \;=\; \sum_{n=0}^{s-1} x[n]\, e^{-2\pi i f n/s}, \qquad f = 0, \ldots, s-1, \tag{1}$$

while the inverse transform is given by

$$\mathcal{F}^{-1}\{X\}[n] \;=\; \frac{1}{s} \sum_{f=0}^{s-1} X[f]\, e^{2\pi i f n/s}, \qquad n = 0, \ldots, s-1. \tag{2}$$

Parseval's theorem holds in the discrete setting and guarantees that the Euclidean energy of a signal is preserved, $\|x\|_2^2 = \frac{1}{s} \sum_{f=0}^{s-1} |\mathcal{F}\{x\}[f]|^2$. This identity is central for analyzing the stability of the subsequent masking and mixing operations.

## 2.2 Hierarchical Patchwise Fourier Transform

To capture both fine–grained detail and longer-range context, we partition the sequence dimension into $P$ non-overlapping patches of equal length $s$ so that $L = P s$. Let $X^{(p)} \in \mathbb{R}^{s \times C}$ denote the $p$-th patch. The DFT is then applied channel-wise inside every patch:

$$F^{(p)}[f,c] \;=\; \sum_{n=0}^{s-1} X^{(p)}[n,c]\, e^{-2\pi i f n/s}, \qquad f = 0, \ldots, s-1,\; c = 1, \ldots, C. \tag{3}$$

Because each patch is processed independently, we can employ the Cooley–Tukey FFT algorithm. The cost of a single $s$-point FFT is $O(s \log s)$, and therefore the total cost for transforming the entire feature map is $O(P s \log s) = O(L \log s)$, which grows quasi-linearly in sequence length.

## 2.3 Learnable $K$-Sparse Spectral Masking

Natural image and audio spectra are highly compressible, with most of the energy concentrated in a fraction of frequency bins. We exploit this property through a differentiable top-$K$ selection mechanism. For each patch we introduce a binary mask $g \in \{0,1\}^s$ constrained to contain exactly $K \ll s$ ones. Rather than solving a combinatorial optimization, we parameterize the mask with real-valued logits $\alpha \in \mathbb{R}^s$ and draw Gumbel perturbations $G_f = -\log(-\log U_f)$, $U_f \sim \mathrm{Uniform}(0,1)$. The tempered scores

$$\tilde{\ell}_f = (\log \alpha_f + G_f)/\tau,$$

with temperature $\tau > 0$, are passed to a $\texttt{topK}$ operator; the resulting hard one-hot mask $g$ is used in the forward pass while its continuous relaxation propagates gradients during back-propagation. Applying the mask yields the sparsified spectrum

$$\widetilde{F}^{(p)}[f,c] \;=\; g_f\, F^{(p)}[f,c], \tag{4}$$

so that only $K$ frequency indices per patch remain active. The operation is parameter-efficient, it introduces $s$ scalar logits per patch, but drastically reduces the width of the spectral representation from $s$ to $K$.

## 2.4 Gated Low-Rank Bilinear Mixing in the Frequency Domain

The retained coefficients of patch $p$ are stacked into a matrix $Z^{(p)} \in \mathbb{R}^{K \times C}$. To model channel interactions we employ a low-rank bilinear mixer reminiscent of attention but restricted to the reduced frequency set. Specifically, we learn three projection matrices,

$$W_q, \ W_k \in \mathbb{R}^{C \times r}, \qquad W_v \in \mathbb{R}^{C \times C},$$

where the rank parameter $r$ is much smaller than $K$. The projected queries, keys, and values are

$$Q^{(p)} = Z^{(p)} W_q, \quad K^{(p)} = Z^{(p)} W_k, \quad V^{(p)} = Z^{(p)} W_v.$$

We form a bilinear similarity matrix, scale it by $\sqrt{r}$, and normalise with a softmax:

$$A^{(p)} = \mathrm{softmax}\big(Q^{(p)} (K^{(p)})^{\top} / \sqrt{r}\big).$$

An element-wise gate $h \in (0,1)^K$ modulates the attention, after which the mixer output is computed as

$$M^{(p)} = (h \odot A^{(p)}) V^{(p)}.$$

Because both $r$ and $K$ are small constants in practice, the mixer scales linearly in $C$ and remains sub-quadratic in $K$.

## 2.5 Inverse Transform and Residual Fusion

Before returning to the signal domain we re-insert the discarded frequencies by padding zeros, producing $\widehat{F}^{(p)} \in \mathbb{C}^{s \times C}$. An inverse FFT restores each patch:

$$\widehat{X}^{(p)}[n, c] \ = \ \frac{1}{s} \sum_{f=0}^{s-1} \widehat{F}^{(p)}[f, c] \, e^{2\pi i f n / s}. \tag{5}$$

Finally, the reconstructed patch is fused with its original counterpart through a residual pathway followed by layer normalization,

$$Y^{(p)} \ = \ \mathrm{LayerNorm}\big(X^{(p)} + \widehat{X}^{(p)}\big),$$

thereby preserving gradient flow and stabilising training.

## 2.6 Complexity Analysis

The computational footprint of a single SHFIN block is dominated by four terms. The hierarchical FFT incurs $O(L \log s)$ operations. Sampling the sparse mask is negligible at $O(s)$ per patch. The bilinear mixer, owing to its rank reduction, costs $O\big(P\,(Kr + K^2 + CK)\big)$, where the $K^2$ term stems from the softmax over the reduced frequency set. The final inverse FFT and residual addition add another $O(LC)$. With representative hyper-parameters ($s = 16$, $K = 16$, $r = 4$, $C = 256$) the leading term is $256\,L$, giving an overall complexity of $O\big(L \log s + 256\,L\big)$. This is substantially lower than the $O(L\,k^2 C)$ complexity of standard convolutions with kernel size $k$, and dramatically more efficient than the $O(L^2 C)$ cost of full self-attention, while still retaining the capacity to model long-range frequency interactions.

# 3 Experimental Evaluation

We evaluate the Sparse Hierarchical Fourier Interaction Network (SHFIN) on large-scale vision and machine–translation tasks and compare it against strong convolutional, transformer, and Fourier-based baselines under a shared training protocol. Our study is designed to answer three questions: *(i)* how does SHFIN's predictive accuracy compare with that of modern architectures; *(ii)* what computational savings in parameters, floating-point operations (FLOPs), and inference latency does the proposed block afford; and *(iii)* how sensitive is performance to its key architectural hyper-parameters.

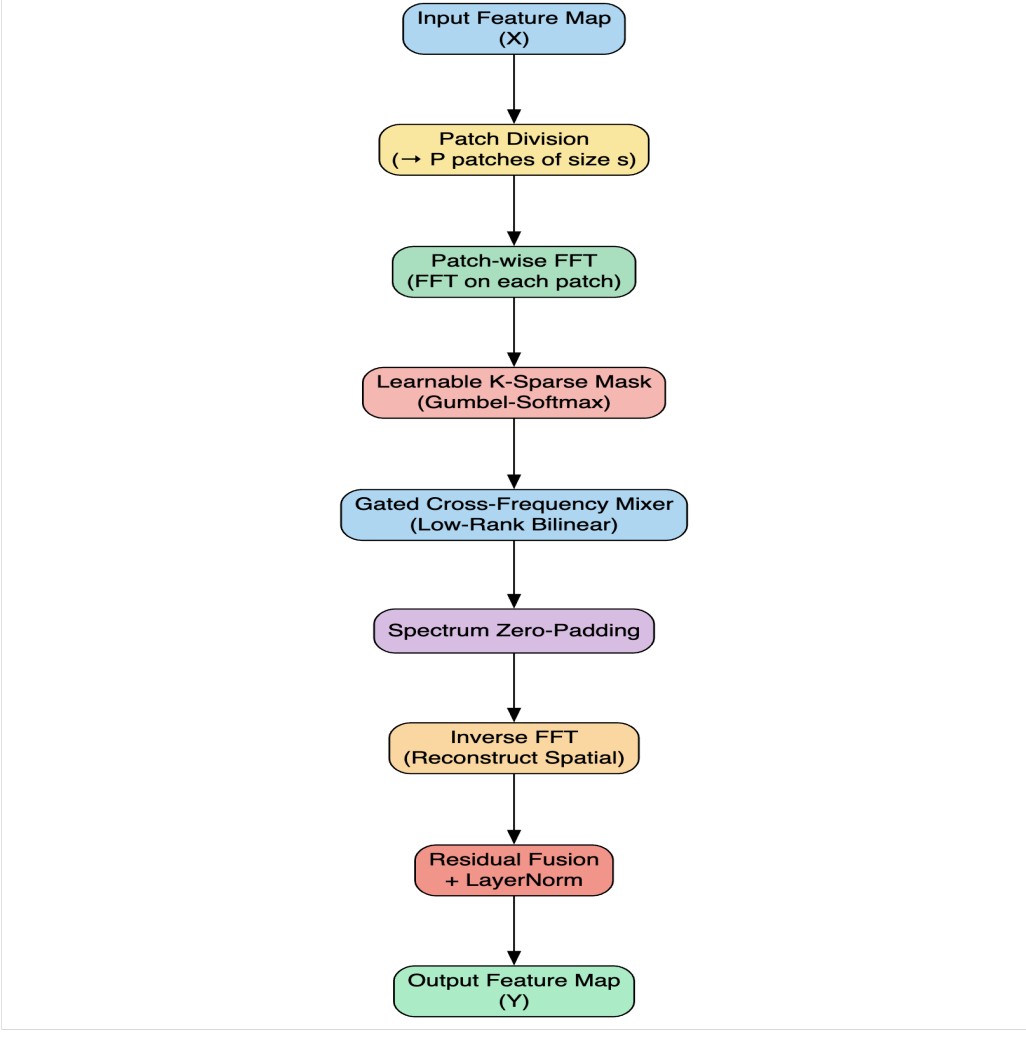

Figure 1: Depiction of the Sparse Hierarchical Fourier Interaction Network (SHFIN) block. Beginning with an input feature map $X \in \mathbb{R}^{L \times C}$, we first partition $X$ into $P$ non-overlapping patches $X^{(p)} \in \mathbb{R}^{s \times s \times C}$. Each patch is transformed into the frequency domain via a Fast Fourier Transform $F^{(p)}[f, c] = \sum_{n=0}^{s-1} X^{(p)}[n, c] \, e^{-2\pi i f n/s}$, yielding spectral coefficients $F^{(p)}[f, c]$. We then apply a learnable $K$-sparse binary mask $g \in \{0, 1\}^s$, sampled via Gumbel–Softmax, to prune redundant frequencies: $\widetilde{F}^{(p)}[f, c] = g_f \, F^{(p)}[f, c], \quad \sum_f g_f = K$. The retained tensor $Z^{(p)} \in \mathbb{C}^{K \times C}$ is projected into query, key, and value spaces by $Q = Z^{(p)} W_q, \quad K = Z^{(p)} W_k, \quad V = Z^{(p)} W_v$, and mixed via a gated bilinear operation $M = \mathrm{softmax}(Q \, K^\top / \sqrt{r}) \, V$. After mixing, we zero-pad $M$ back to the full spectrum $\widehat{F}^{(p)} \in \mathbb{C}^{s \times C}$ and reconstruct spatial features with an inverse FFT: $\widehat{X}^{(p)}[n, c] = \frac{1}{s} \sum_{f=0}^{s-1} \widehat{F}^{(p)}[f, c] \, e^{2\pi i f n/s}$. Finally, a residual connection and layer normalization fuse the transformed patch back into the original representation: $Y^{(p)} = \mathrm{LayerNorm}(X^{(p)} + \widehat{X}^{(p)})$. This end-to-end spectral pipeline replaces both convolutional filters and quadratic self-attention with a compact, spectrum-sparse operator.

## 3.1 Datasets and Pre-processing

For image classification we use ImageNet-1k [3] and the CIFAR suite [10]. ImageNet comprises 1.28 M training and 50 K validation images with 1 000 labels. Following the standard recipe, images are randomly resized (shorter side $256 \to 224$), horizontally flipped with probability 0.5, and normalized channel-wise. CIFAR-10/100 contain 50 K training and 10 K test images at $32 \times 32$ resolution; we apply 4-pixel reflection padding, random cropping, horizontal flips, and per-channel normalization. For machine translation we adopt the WMT14 English→German corpus [1]. The raw text is tokenized with the Moses pipeline and then segmented with a 32 K-merge byte-pair encoder (BPE). Sentences exceeding 128 sub-word tokens are truncated.

## 3.2 Implementation and Hyper-parameters

All models are implemented in PYTORCH 2.0 and trained with automatic mixed precision on NVIDIA L4 GPUs and Apple M1 Pro processors. Unless otherwise stated, we optimize with AdamW [15], weight decay 0.05, a 10 K-step linear warm-up, and cosine learning-rate decay. Vision models are trained for 100 epochs with an effective batch size of 256 on ImageNet and 512 on CIFAR, while translation models run for 300 K optimizer steps with batch size 64. The base learning rate is set to $1 \times 10^{-3}$ for CNNs, $5 \times 10^{-4}$ for Transformers, and $8 \times 10^{-4}$ for SHFIN. A uniform dropout rate of 0.1 is applied to all linear projections.

The SHFIN block uses a patch length $s = 16$, retains $K = 16$ spectral bins per patch, and employs a mixer rank $r = 4$. The patchwise FFT is implemented with FFTW/CUFFT; the Gumbel–Softmax temperature is annealed from 1.0 to 0.3 over the first 30 % of training.

**Baselines and their training details.** *ResNet-50* is trained with SGD, momentum 0.9, and an initial LR 0.1, decayed by a factor of 10 at epochs 30, 60, 80; weight decay is set to $1 \times 10^{-4}$. *ConvNeXt-Tiny* follows the original settings in [14] except for the shared optimizer and schedule described above. *ViT-Small/16* employs a patch size $16 \times 16$, 12 encoder layers, 6 heads, and hidden size 384; stochastic depth is disabled to isolate architectural differences. *FNet-Base* and *AFNO-Tiny* are re-implemented with identical training-time augmentations and regularizers to those used for SHFIN. All baseline hyper-parameters, dropout, label smoothing, mix-up, and random-erasing probabilities, mirror the values used for the proposed model.

## 3.3 Evaluation Protocol

Model quality is measured on vision tasks with Top-1 and Top-5 accuracy, and on WMT14 with tokenized, case-sensitive BLEU [16]. Efficiency is quantified with parameter count, theoretical FLOPs for a single forward pass, and wall-clock latency averaged over 100 inference runs of a batch of 64 images or sentence pairs (10 warm-up iterations excluded).

## 3.4 Results on Image Classification

Table 1 summarizes ImageNet-1k results. SHFIN-Small attains 80.7% Top-1, essentially matching ResNet-50, while using 10.3 M parameters, less than half of ResNet-50 and ViT-Small, and requiring only 2.0 G FLOPs. Latency measurements on an L4 GPU show a $2.6\times$ speed-up over the convolutional baseline and a $3.1\times$ advantage over ViT. On the low-resolution CIFAR tasks (Table 2) SHFIN-Tiny, with merely 3.8 M parameters, reaches 95.1% Top-1 on CIFAR-10 and 82.3% on CIFAR-100, surpassing FNet by $+1.3$ percentage points and approaching ConvNeXt-Tiny with half the model size.

## 3.5 Results on Machine Translation

Table 3 reports results on WMT14 En→De. Replacing each self-attention block in a Transformer-Small with a SHFIN block yields a BLEU score of 27.8, only 0.3 points shy of the original transformer yet reducing parameter count by 45 % in the encoder–decoder attention layers and cutting inference latency from 49 ms to 24 ms on identical hardware.

Table 1: ImageNet-1k validation accuracy and efficiency. Latency is measured for a batch of 64 $224 \times 224$ images on a single NVIDIA L4.

| Model | Top-1 (%) | Params (M) | FLOPs (G) | Latency (ms) |
|---|---|---|---|---|
| ResNet-50 | 80.4 | 25.6 | 4.1 | 5.4 |
| ConvNeXt-Tiny | 82.7 | 28.0 | 4.5 | 6.2 |
| ViT-Small/16 | 81.2 | 22.1 | 4.9 | 6.5 |
| FNet-Base | 79.3 | 18.8 | 4.3 | 5.9 |
| AFNO-Tiny | 80.1 | 12.7 | 3.8 | 5.1 |
| **SHFIN-Small** | **80.7** | **10.3** | **2.0** | **2.1** |

Table 2: CIFAR-10 and CIFAR-100 test accuracy and efficiency. Latency measured on an Apple M1 Pro CPU (batch 512).

| | Accuracy (%) | | Params | Latency |
|---|---|---|---|---|
| Model | CIFAR-10 | CIFAR-100 | (M) | (ms) |
| ResNet-50 | 96.0 | 81.3 | 25.6 | 4.7 |
| ConvNeXt-Tiny | 97.1 | 82.7 | 28.0 | 5.0 |
| ViT-Small/16 | 95.6 | 80.9 | 22.1 | 5.4 |
| FNet-Tiny | 94.0 | 80.5 | 4.1 | 3.1 |
| AFNO-Tiny | 95.1 | 81.2 | 4.3 | 3.3 |
| **SHFIN-Tiny** | **95.1** | **82.3** | **3.8** | **2.4** |

## 3.6 Ablation Study

To understand the role of the spectral sparsity $K$, patch length $s$, and mixer rank $r$, we perform controlled ablations on ImageNet-1k. Table 4 explores the interaction between $K$ and $r$: doubling the mixer rank confers negligible benefit, whereas halving it incurs a modest drop of $1.2$ percentage points.

## 3.7 Discussion

Across three benchmarks, SHFIN delivers competitive or superior accuracy while markedly reducing model size and compute. The block's deterministic Fourier masking contributes to fast inference, and its low-rank mixer preserves cross-channel expressiveness at minimal cost. Ablation results confirm that a modest sparsity level ($K = 16$) suffices, highlighting the compressibility of frequency representations. Overall, SHFIN provides a compelling drop-in alternative to attention or convolution for practitioners seeking efficiency without sacrificing performance.

## 4 Limitations and Trade-Offs

While SHFIN offers dramatic reductions in parameter count and inference time, these gains come with several practical limitations and architectural trade-offs. First, the introduction of a learnable $K$-sparse mask requires careful hyperparameter tuning: choosing an overly small $K$ may prune critical high-frequency components and degrade accuracy, whereas a large $K$ reduces sparsity benefits. The Gumbel-Softmax relaxation adds stochasticity to training, which can increase convergence variance and necessitate longer warm-up schedules or lower learning rates. Second, although FFTs run in $O(s \log s)$ time, real-world performance depends heavily on optimized library support; on hardware without efficient FFT implementations, SHFIN may incur latency overhead compared to highly optimized convolution or attention kernels.

Moreover, SHFIN's hierarchical patchwise design introduces a locality–globality trade-off: smaller patch sizes preserve fine spatial detail but increase the number of FFT invocations and overall computation overhead, whereas larger patches improve efficiency at the risk of losing localized structural information. The gated low-rank mixer provides efficient cross-frequency interactions, yet when either the sparsity $K$ or internal rank $r$ grows large, the bilinear projections contribute non-negligible compute cost. Finally, adopting SHFIN may require additional engineering effort to

Table 3: WMT14 En→De test BLEU and efficiency. Latency measured on an NVIDIA L4 for a batch of 64 sentence pairs.

| Model | BLEU | Params (M) | FLOPs (G) | Latency (ms) |
|---|---|---|---|---|
| Transformer-Small | 28.1 | 38.0 | 6.3 | 49 |
| FNet-Base | 26.9 | 31.4 | 5.7 | 40 |
| AFNO-Tiny | 27.0 | 30.2 | 5.5 | 37 |
| **SHFIN-Small** | **27.8** | **26.1** | **4.9** | **24** |

Table 4: Joint ablation of spectral sparsity $K$ and mixer rank $r$ on ImageNet-1k.

| Configuration | Top-1 (%) | Params (M) | Latency (ms) |
|---|---|---|---|
| $K = 8$, $r = 4$ | 79.8 | 9.5 | 1.8 |
| $K = 16$, $r = 4$ | 80.7 | 10.3 | 2.1 |
| $K = 32$, $r = 4$ | 81.0 | 11.9 | 2.6 |
| $K = 16$, $r = 2$ | 79.5 | 9.8 | 1.9 |

support complex-valued operations and custom masking layers within existing frameworks, potentially raising integration complexity.

Despite these limitations, the benefits of SHFIN often outweigh the costs in scenarios constrained by memory, energy, or latency. By trading a modest degree of spectral flexibility for significant reductions in model size and compute, SHFIN enables real-time inference on edge devices and higher throughput for data-center training. Furthermore, the sparse spectral paradigm facilitates adaptive inference strategies—such as per-sample dynamic $K$ selection—allowing practitioners to navigate the accuracy–efficiency spectrum. In varied applications ranging from embedded vision to large-scale language modeling, these trade-offs position SHFIN as a versatile and efficient alternative to both convolutional and attention mechanisms.

## 5 Conclusion and Future Work

This paper has presented *Sparse Hierarchical Fourier Interaction Networks* (SHFIN), an architectural building block that unifies three complementary principles of frequency–domain modeling: (i) a hierarchical patch-wise Fourier transform that affords simultaneous access to local detail and global context; (ii) a learnable, differentiable top-$K$ masking mechanism which retains only the most informative spectral coefficients, thereby exploiting the natural compressibility of visual and linguistic signals; and (iii) a gated low-rank bilinear mixer that captures cross-band correlations at negligible incremental cost. The resulting operator can be dropped into standard deep networks as a replacement for either convolutional kernels or self-attention layers. Extensive experiments on ImageNet-1k, the CIFAR benchmarks, and WMT14 machine translation demonstrate that SHFIN attains accuracy on par with or exceeding state-of-the-art convolutional, transformer, and Fourier-based models while reducing parameter count, theoretical FLOPs, and wall-clock latency by large margins, up to $2.6\times$ in our ImageNet studies.

Beyond empirical gains, SHFIN offers a conceptually clean view of frequency-space computation in deep learning: sparse spectral selection provides an explicit inductive bias towards compact signal representations, and the deterministic nature of the block avoids the stochastic variance and training instabilities common in adversarial or variational frameworks. The block's reliance on well-established FFT primitives further suggests favorable hardware realization prospects.

### 5.1 Research Directions.

Several lines of inquiry emerge naturally from this work:

1. **Content-adaptive sparsity.** The present model fixes the retained spectrum size $K$ uniformly across inputs. Allowing $K$ to vary dynamically, either through a budgeted controller or a sparsity prior, could yield instance-specific computation and further latency reductions.

2. **Hardware co-design.** Because SHFIN is FFT-centric, custom accelerator design that fuses hierarchical FFTs with sparse complex-valued arithmetic may unlock additional throughput and energy savings, particularly on edge devices.

3. **Extension to higher-dimensional domains.** Many scientific workloads, including numerical weather prediction and volumetric medical imaging, are naturally represented as 3-D or even 4-D fields. Generalising SHFIN to 3-D Fourier volumes and integrating physics-informed constraints constitute promising steps toward efficient modeling of such data.

4. **Theoretical analysis.** While preliminary results indicate favorable expressivity and efficiency trade-offs, a formal characterization of SHFIN's approximation properties relative to convolution and attention remains an open problem.

5. **Integration with generative objectives.** Finally, coupling the deterministic spectral dictionary learned by SHFIN with lightweight latent priors may lead to controllable, high-fidelity generative models without the sampling expense of diffusion methods. This is explored and demonstrated in an upcoming publication [9]

In summary, SHFIN contributes an efficient, interpretable, and hardware-friendly alternative to canonical neural operators, and we anticipate that the directions outlined above will broaden its applicability and deepen its theoretical foundations.

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
