# OpenReview forum: "Reducing Deep Network Complexity via Sparse Hierarchical Fourier Interaction Networks"
_NeurIPS.cc/2025/Conference — Submitted to NeurIPS 2025_

### Official Review · Reviewer_vkhe · 2025-06-10

**Clarity:** 2
**Significance:** 2
**Originality:** 2
**Rating:** 2
**Confidence:** 3

**Summary:**

This paper proposes a drop-in replacement of vanilla convolution and self-attention layer through the lens of the Fourier transform. It introduces a patch-wise FFT, a mask to select key frequency components, and a low-rank mixer, together resulting in better performance and reduced space/time complexity.

**Questions:**

1. What is the rationale behind the limitations introduced from line 59? Is the significance sufficiently supported or widely acknowledged? Are there any prior arts that relate to the problems to be addressed by the authors? There should be a section on related work depicting the broad context.
2. Are there any criteria to guide the selection of $K$?
3. Are there any reasons to use gating in line 129?
4. How do the authors adapt the proposed method to convolution? Why is the complexity $O(L \log s +256L) $ lower than convolution $O(L k^2C)$ (how to get it?); aren’t they both linear w.r.t $L$? The comparison to convolution could be misleading if there is no such derivation.

**Ethical Concerns:**

["NO or VERY MINOR ethics concerns only"]

**Final Justification:**

I will keep the current rating.

**Limitations:**

Yes, the authors include limitations in the manuscript. I suggest conducting experiments on a broader range of tasks and open-sourcing the implementations.

**Paper Formatting Concerns:**

I do not see any major formatting issues.

**Quality:**

1

**Strengths And Weaknesses:**

Strengths:
1. The proposed method outperforms Transformer and other frequency-domain architectures with improved parameter and computational efficiency.

Weakness:
1. The limitations of Fourier-based architectures that the authors bring up are somewhat unclear and trivial to me: redundant components, lack of locality, and spectrum sparsity. I am not fully convinced they are the critical issues we need to take good care of.
2. There is no empirical support that the selection of $K$ frequency works as the authors intended, i.e., focusing on the concentrating frequency.
3. The ablation experiments are not strong enough. There are unconventional add-ons or alternatives to be investigated, e.g., gating in the attention, strategies to select $K$ frequency components.
4. Despite promising results, though limited, I am a little concerned about the reproducibility and the technical soundness. In particular, I am quite skeptical about how much information the query and key can store when compressed to such a small size $K \times r$, which is $16 \times 4$ according to hyperparameters in line 144.

---

### Official Review · Reviewer_M6es · 2025-06-26

**Clarity:** 2
**Significance:** 3
**Originality:** 2
**Rating:** 2
**Confidence:** 4

**Summary:**

This research presents Sparse Hierarchical Fourier Interaction Networks (SHFIN), an innovative architectural component intended to function as a cohesive and efficient substitute for convolutional kernels in CNNs and self-attention mechanisms in Transformers. SHFIN's fundamental concept involves executing computations in the frequency domain through a three-stage methodology: 1) a hierarchical, patch-wise Fast Fourier Transform (FFT) to encompass both local and global contexts; 2) a learnable K-sparse masking mechanism, realized via Gumbel-Softmax, to identify the most salient frequency components; and 3) a gated, low-rank bilinear mixer to facilitate efficient cross-channel interactions within the sparse spectral subspace. The authors demonstrate through experiments on image classification (ImageNet, CIFAR) and machine translation (WMT14) that SHFIN-based models attain performance comparable to robust baselines while markedly decreasing parameter count, FLOPs, and inference latency.

**Questions:**

1) Would you kindly share more details about the SHFIN block's design philosophy? Why was this particular order of actions selected? Why, for example, is the cross-frequency mixer positioned after the sparsity mask? Justifying the importance and contribution of each design feature would need a more complete ablation research in which essential SHFIN block components are systematically removed or altered (for example, by doing an experiment without the Gumbel-Softmax mask or the low-rank projection).

2) Would you kindly explain the precise building of a CNN based on SHFIN? Was a ResNet-50 model explicitly adjusted using SHFIN blocks for the ImageNet tests, and if so, how? The paper's key assertion that the SHFIN-CNN is a universal operator would be considerably bolstered if thorough results were supplied for the model on a benchmark such as ImageNet.


3) Could you expand on why the performance metrics in Tables 1 and 2 are bolded?  The current formatting is ambiguous because SHFIN's results are not the best in a number of critical categories (such as Top-1 accuracy). To guarantee clarity and a fair comparison, I would advise adopting the customary method of bolding only the best result per column.

**Ethical Concerns:**

["NO or VERY MINOR ethics concerns only"]

**Limitations:**

Yes.

**Paper Formatting Concerns:**

No major formatting issues were found in this paper.

**Quality:**

2

**Strengths And Weaknesses:**

Strengths:

The study focuses on developing a unified operator that can effectively replace both self-attention mechanisms and convolutional operations—an advancement that holds significant potential for streamlining neural network architecture design.

Weaknesses:
1) The report does not fully justify the unique design decisions taken for the SHFIN block. Why this exact component order was picked over alternate configurations is unknown. The presented ablation study is not a genuine architectural ablation, but rather a hyperparameter sensitivity analysis (changing K and r). To highlight the significance of each component, a good ablation would require removing or changing out individual SHFIN block components (for example, the gating mechanism or the bilinear mixer for a more simplistic MLP).

2) The paper's explanation of the SHFIN block's incorporation into comprehensive network designs is unclear. The building of a SHFIN-based CNN is not explained, despite the fact that SHFIN-Small is built on Transformer-Small. For example, are particular convolutional layers or all of them replaced? The argument that SHFIN is a universal substitute for convolution is restricted by the apparent absence of experimental data for a CNN backbone updated with SHFIN on ImageNet.  It would be more intriguing to see SHFIN utilized with other contemporary and different backbones, such as DenseNet or EfficientNet, as the tests are limited to a small number of base models (ResNet, ViT).

3) The tables' results presentation may be deceiving. Although it is not the highest value in the column (ConvNeXt-Tiny at 82.7%), the Top-1 accuracy for SHFIN-Small (80.7%) is bolded in Table 1. Table 2 indicates a similar problem for both the CIFAR-10 and CIFAR-100 results. The best-performing result in each column is not correctly bolded. This presenting decision might mislead readers and misrepresent how well the model performs in compared to its counterparts.

4) There aren't many baseline models, and they're a little out of date. Even though ViT-Small and ResNet-50 are industry standards, the field of efficient architectures is evolving swiftly. A more realistic evaluation of SHFIN's position in the present landscape would come from comparison with more contemporary and pertinent models, particularly other effective Fourier-based or state-space models (e.g., S4, Mamba).

---

### Official Review · Reviewer_C1oM · 2025-07-02

**Clarity:** 1
**Significance:** 2
**Originality:** 1
**Rating:** 1
**Confidence:** 4

**Summary:**

The paper proposes Sparse Hierarchical Fourier Interaction Networks (SHFIN), which perform convolution and attention operations in the spectral domain, reducing the parameter count and computational complexity. It also incorporates selective mixing by exploiting sparsity in the Fourier domain, induced by learnable K-sparse frequency masking that prunes redundant bands.

**Questions:**

1. What is the backbone architecture of SHFIN? Is it ResNet-50 and Transformer-small for vision and translation tasks, respectively?
2. What are the FLOPs and model parameters breakdown among different modules (convolution, pointwise convolution, fully connected layer) in the baseline architecture?
3. What are the FLOPs and model parameters gains from spatial to spectral domain conversion (FFT) and from frequency pruning?
4. Are all evaluations carried out on fp32 quantization format? What do you mean by “automatic mixed precision” in line 166?
5. Are the weights of the SHFIN complex-valued? If so, is a complex-valued parameter consisting of a real and imaginary part counted as a single parameter in Table 1-4?

**Ethical Concerns:**

["NO or VERY MINOR ethics concerns only"]

**Final Justification:**

The authors did not participate in the discussion without any rebuttal. The concerns remain as is -- I will maintain my rating.

**Limitations:**

Yes - the paper devotes significant space to disclose its limitations in Section 4.

**Paper Formatting Concerns:**

- Inconsisent definitions: $Z^{(p)} \in \mathbb{C}^{K \times C}$ in Fig.1 on page 5, whereas $Z^{(p)} \in \mathbb{R}^{K \times C}$ in line 124.
- Inconsisent definitions: Patch $X^{(p)} \in \mathbb{R}^{s \times C}$ in line 105, whereas $X^{(p)} \in \mathbb{R}^{s \times s \times C}$ in Fig.1 on page 5.
- Typo(?): FNet-Tiny in Table 2 should be FNet-base if consistent with Tables 1 and 3.

**Quality:**

2

**Strengths And Weaknesses:**

Strength

- The paper has a somewhat clear direction to reduce computational complexity and model size.
- Using learnable masking in the spectral domain, realized by Gumbel-softmax, is interesting.
- There seems to be performance gains against the baseline models and other spectral domain counterparts (e.g., FNet and AFNO).

Weakness

- The novelty of this work is not clear — the use of DFT to reduce computational complexity is already heavily explored in the past and frequency masking was already proposed in AFNO.
- The experimental evaluation is limited; in order to validate the efficacy of SHFIN, more baseline models should be incorporated. More models like ViT, MobileNet, EfficientNet, or BERT, other than ResNet-50 and transformer-small, should be tested to validate the effectiveness of SHFIN.
- The paper is not very clear in clarifying the model architecture. (See question 1)
- The work does not elaborate on the source of model parameter reduction. (See questions 2 and 3)
- While the authors claim the framework is “generalized,” it does not accelerate the fully-connected layers in convnet or feedforward layers in transformers, thereby limiting the acceleration opportunity.
- Section 2 mixes up mathematical formulations of replacing convolution and attention with SHFIN, making readers rather confused.
- Not sure if the complexity analysis in Section 2.6 is sound. For example, it claims $O(L\log s+ 256L)$ compared to $O(Lk^2C)$… which should be $k^2=9$ times faster given $C=256$ in line 146. Why is it so? Perhaps because the operations are bottlenecked by pointwise convolutions?

---

### Official Review · Reviewer_12mi · 2025-07-04

**Clarity:** 2
**Significance:** 1
**Originality:** 1
**Rating:** 1
**Confidence:** 5

**Summary:**

This paper proposes a lightweight neural architecture for image-based tasks by leveraging modeling in the Fourier domain. In the proposed framework, input images are first divided into patches, and each patch is transformed into the Fourier space. A learnable sparse spectral mask is then applied to select a small subset of frequency bands, effectively reducing the dimensionality and focusing on informative components. The filtered spectral representations are subsequently processed using a low-rank attention across the sequence of patches. Finally, the processed signals are transformed back to the spatial domain via the inverse Fourier transform.

**Questions:**

N/A

**Ethical Concerns:**

["NO or VERY MINOR ethics concerns only"]

**Limitations:**

Some limitations are mentioned in Sec. 4. But from my perspective, this paper is limited in novelty, core contribution, and empirical soundness.

**Paper Formatting Concerns:**

Fig. 1 takes one page.
There is a list of keywords under the abstract, which is unusual.

**Quality:**

1

**Strengths And Weaknesses:**

**Strengths**

+ The proposed architecture is simple and easy to follow. The writing is clear and avoids whistles and bells.

**Weaknesses**

- The technical novelty is limited. The idea of processing images in the Fourier domain is well-explored in prior literature. Moreover, the use of spectral masking and low-rank attention appears to be ad hoc; the paper lacks a clear explanation of their motivation or theoretical principles. Simply reducing the intermediate dimension in attention is a common practice and not, in itself, a novel contribution.

- The experimental evaluation is insufficient. The proposed model is compared to only a small set of baselines, and even in these limited comparisons, it underperforms relative to standard methods. While the paper highlights lower latency, this comes at the cost of degraded accuracy. Additionally, experiments are conducted only on small-scale settings.

- The presentation and formatting are poor. The full paper spans only 8.5 pages, and a significant portion is devoted to low-density content. For example, Sec. 4 and 5 together take up just two pages talking about limitation and future directions, Fig. 1 occupies an entire page unnecessarily, and a basic introduction to the Fourier transform consumes half a page. This sparsity gives me the impression of underdeveloped content.

---

### Decision · Program_Chairs · 2025-09-17

**Decision:**

Reject

**Comment:**

The paper proposes a sparse hierarchical fourier interaction networks (SHFIN) to perform convolution operations in the spectral domain to reduce computational complexity. However, both the methodological novelty and the ablation studies do not meet the expected standards for a NeurIPS publication.